# Impact of changes in chatbot's facial expressions on user attention and reaction time

**Kamil Bortko****\*, Kacper Fornalczyk, Jarosław Jankowski, Piotr Sulikowski****, Karina Dziedziak**

Faculty of Computer Science and Information Technology, West Pomeranian University of Technology, Szczecin, Poland

\* kbortko@zut.edu.pl

## Abstract

Communication within online platforms supported by chatbots requires algorithms, language processing methods, and an effective visual representation. These are crucial elements for increasing user engagement and making communication more akin to natural conversation. Chatbots compete with other graphic elements within websites or applications, and thus attracting a user's attention is a challenge even before the actual conversation begins. A chatbot may remain unnoticed even with sophisticated techniques at play. Drawing attention to the chatbot area localized within the periphery area can be carried out with the use of various visual characteristics. The presented study analyzed the impact of changes in a chatbot's emotional expressions on user reaction. The aim of this study was to observe, based on user reaction times, whether changes in a chatbot's emotional expressions make it more noticeable. The results showed that users are more sensitive to positive emotions within chatbots, as positive facial expressions were noticed more quickly than negative ones.

## Introduction

Chatbots are widely used within online platforms. They are based on human intellectual principles and abilities that focus on conversation. An organization can benefit from a chatbot's unique knowledge based on consumer opinions in addition to its human-computer interaction (HCI) area background, allowing for the redistribution of cognitive tasks between humans and machines [1]. Chatbots integrate many characteristics, making them capable of eliciting different kinds of social responses to varying degrees, including verbal responses [2], gestures, and visual responses [3].

While language processing, machine learning, and artificial intelligence algorithms supporting communication within chatbots have been extensively analyzed [4, 5], only a limited number of studies have focused on their visual parameters [6]. Less focus has been given to the appearance of chatbots, including elements to attract a user's eyesight and attention, but the overall impact of chatbot design has been emphasized [7]. The growing popularity of chatbots has brought about new challenges regarding HCIs such as changing interaction patterns [8],

**Data Availability Statement:** data is here https:// github.com/kbzut/Periperial-chatbot-faces.

**Funding:** This work was supported by the National Science Centre of Poland, Decision No. 2017/27/B/ HS4/01216 (J.J., K.B.) and this research was

supported by ZUT Highfliers School /Szkoła Orłów ZUT/ project co-ordinated by Dr. Piotr Sulikowski, within the framework of the program of the Minister of Education and Science /Grant No. MNiSW/2019/391/DIR/KH, POWR.03.01.00-00-P015/18/, co-financed by the European Social Fund, the amount of financing PLN 1,704,201,66 (K.F.). The funders had no role in study design, data collection and analysis, decision to publish, or preparation of the manuscript.

**Competing interests:** The authors have declared that no competing interests exist.

and thus, the role of chatbot usability has also been discussed [9]. The need to further improve their performance is due to raising expectations that chatbots will display social behaviors that are customary for interpersonal communication.

The extent to which a medium, like a chatbot, is designed to resemble and behave like a human, incorporating elements such as a human-like appearance, facial expressions, and gestures, can significantly influence the perceived level of social presence during the interaction. Consequently, users may experience a heightened sense of engaging in a genuine and natural conversation, almost as if they were conversing with an actual human being rather than a machine.

The visualization of chatbots and their social features should target users' expectations to ultimately avoid frustration and dissatisfaction. The effects of an electronic conversation on human behavior and the perceived level of anthropomorphism are a part of the broad issue of human attitudes towards humanoid technologies.

According to the theory proposed by Mori, the more a character resembles a human, the more it is accepted by us and evokes positive feelings. However, when a character becomes too realistic but still has subtle differences, such as unnatural movements or improper proportions, we experience a sense of unease or rejection in our minds. This is the moment when we enter the so-called Uncanny Valley [10, 11].

Mori's theory suggests that the acceptance of artificial figures increases with their level of realism until a certain point, after which there is a sudden decline in acceptance. Only when a character reaches an exceptionally high level of realism, almost indistinguishable from a living human, does acceptance increase again.

The phenomenon of the Uncanny Valley has also been applied in the context of avatars with facial expressions. Irregularities in facial expression movements or inconsistencies with our expectations can make us feel uneasy and result in negative attitudes towards such avatars. Therefore, it is not surprising that the hypothesis regarding the Uncanny Valley has been adopted to explain the poor commercial success of some animated films in the media [12].

In a study conducted by Katsyri et al. (2015), existing research on people's reactions to artificial figures with varying degrees of realism was analyzed to investigate which hypotheses best explain this phenomenon. One hypothesis suggests that the feeling of unease in the Uncanny Valley arises from our social and cultural context. If artificial human-like figures are perceived as strange or inappropriate in our society, they can evoke negative emotions [13]. Study of Kao (2019) finds that avatars with higher anthropomorphism led to higher player experience. Avatars with higher anthropomorphism also led players to identify more highly with their avatars. Independent of avatar type, we find avatar identification significantly promotes player experience. Players playing games doomed by little humanoidness. We will be more successful when the avatar is more a human [14].

Another factor potentially influencing chatbot design is that in natural communication, participants pay attention to emotions expressed through voice and movements such as gestures and facial [15]. The same mechanisms transferred to a virtual environment can be related to emotions expressed within a chatbot visualization. The content displayed in relation to human perception is intended to arouse feelings and the desire to interact with the chatbot [16]. A conversation with a graphic avatar in the chatbot field [17] will have a different impact on a user than a conversation with a human visualization, and may increase the fluency of conversation. Facial expressions improve the effectiveness of avatars in different contexts [18] and are also a way of influencing people's judgment [17]. Naturalness is important, as evidenced by how well a dialogue system can follow a natural course of conversation. A method of evaluating naturalness in conversational dialog systems has been proposed [19] based on a chatbot that summarizes the user's emotional state in a survey. The percentage of the chatbot's facial

expressions followed by the user during the conversation with the chatbot and the interaction with the content were analyzed. The main aspects studied were the communication between the chatbot and the human, whether the chatbot should be more of an agent, etc. The chatbot's visual area itself, however, leaves room for a deeper analysis [7]. The results of studies on chatbot visualization have emphasized the importance of the visual appearance of the chatbot [20].

The main goal of the presented study was to verify the impact of emotional changes in the chatbot's face on user attention and reaction time. Earlier studies demonstrated the impact of textual content change intensity within the chatbot on user attention [21]. Expressing emotions can be one of the techniques to attract user attention, and the question is which emotional state can effectively attract user attention during other tasks without the risk of there being negative effects on user perception [22]. Chatbots are usually located within corners of the screen and the peripheral vision area. Our experiment integrates a task within the primary vision area and the presence of the chatbot in the peripheral area. The primary task requires the participant's attention, and changes within the chatbot are carried out in the periphery [23]. In real systems, chatbots (as well as system messages or in-video-game messages) that are aimed at helping users or players, are usually located peripherally. Considering that, they require the users to perform a more accurate peripheral search and identification tasks while performing a central task [24]. This is in line with the attention-utility issue discussed by McCrickard et al. [25] in the context of peripheral design, which refers to the ways in which peripheral cues or messages help users achieve their goals without requiring their full attention. Utility refers to a system's usefulness to its users or customers. McCrickard and Chewar discussed utility as a value provided by the peripheral system as a whole, did not directly manipulate utility as part of their experiment, and focused on attention-utility trade-off, considering attention as a constrained resource that can be traded for some utility [26]. We would like to consider utility as the meaning of the content of the "individual gaps", which refer to specific needs or requirements of individual users that may not be fulfilled by the existing attention-utility of the system. By addressing these individual gaps, the authors suggest that the attention-utility of the system can be increased and adapted to better meet the diverse needs of users. Utility can be evaluated as an aspect of human-computer interaction for the purpose of identifying aspects of this interaction that can be improved with the help of evaluation methods [27] in different experimental settings. In studying various peripheral or secondary display information representations, many researchers focus only on the information gained without measuring the changes in the primary task performance caused by these display [27].

Similarly, some studies create an unrealistic testing environment given that the distinction between reaction and comprehension is unclear. While user reaction and comprehension tasks are often closely related, the two objectives may imply differences in the notification system information design.

In everyday life, signals such as facial expressions often appear in our peripheral field of vision. Although the processing of facial expressions within the central vision has been widely researched [28], fewer studies have focused on processing objects within the peripheral vision. To date, research has been consistent about the specific and automatic processing of information about positive emotions in the peripheral vision, which can draw attention to emotion-enhanced messages and enable a quick behavioral response [28]. One study showed a decline in recognition and detection performance as eccentricity increases, with happiness and surprise being the best recognized expressions in the peripheral vision. As for detection, however, another well-detected expression is fear, along with happiness and surprise [29].

Based on the presented experiment, we show that positive emotions, such as happiness and surprise in a human chatbot representation, are better detected in terms of reaction time than other expressions. The results show that task constraints shape the perception of expressions

in the peripheral vision and provide new evidence that detection and recognition rely on partially separate underlying mechanisms, with the latter being more dependent on the higher spatial content frequency of the facial stimulus.

## Experiment setup

The main goal was to study the user's reaction to changes in the emotional state of a chatbot located within the area of the peripheral vision. The structure of the experiment is presented in Fig 1. The screen area was organized into two sections: the center of the screen showing the primary task, and the chatbot area in the periphery. The experiment was set up so that the participants could not look directly at the chatbot due to validation and real-time control of the coordinate position of gaze patterns. The experiment was planned to last 15 minutes per user, and each user session was divided into four parts. In the first part, the user's reaction was examined on the basis of Male Face 1. The reaction was pressing the mouse button when a change in the object (in our task, a photo of a face with an expressed emotion) was observed in the chatbot located in the peripheral area. In the second and third parts, Female Face 1 and 2 were used, and in the fourth part, Male Face 2 was used. For each part of the experiment, the individual six emotions were displayed in the following order: happiness, sadness, anger, fear, surprise, and disgust. Each of these emotions was displayed twice. The appearance of each emotion was preceded by the display of a photo with a neutral expression on the face, which was the so-called zero state. The neutral state separated the emergence of each emotion to reset its impact. The change in the photo in the chatbot area happened after a random time of 5–12 s.

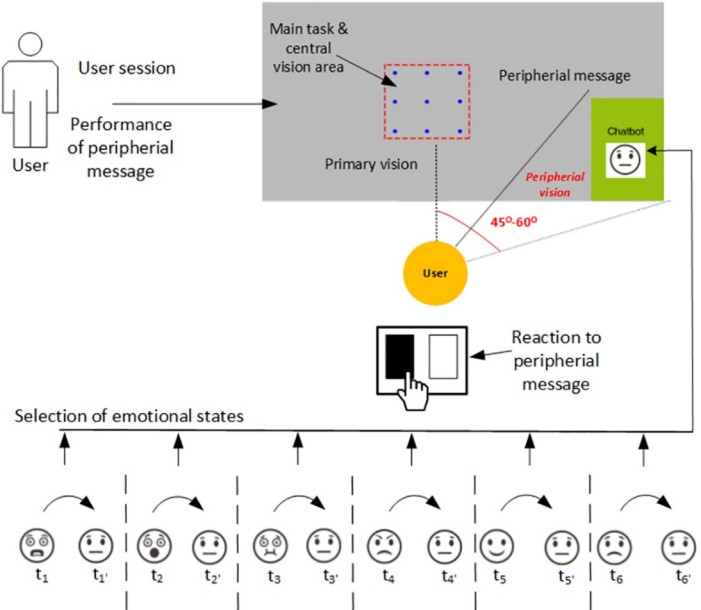

**Fig 1. Each study participant's experimental session was based on a primary task within the central vision area and a chatbot component located in the periphery, in the bottom right-hand corner of the screen.** A user was performing tasks, and different emotions were expressed within the chatbot area. Each change in facial expressions noticed in the chatbot area was reported by the user's mouse left click. Emotions were always changed from a neutral state, and after each change the neutral state returned.

The experiment involved 20 people aged 20-25. Of study participants, 50 per cent were female, and 50 per cent were male. The inclusion criteria were defined as 'participants with normal or corrected-to-normal vision and between 18 and 35 years old'. The exclusion criteria were defined as 'participants with a history of eye disease or neurological disorder, or any other condition that could affect the results of the study. The experiment used a 27-inch Dell monitor and the Tobii Pro X3 eye tracker with a sampling frequency of 120 Hz. A special stand with a tripod was prepared for the experiment, which made it possible to keep the head of each participant in a stationary position. The eye-tracking software used was Tobii Pro Lab, version 1.194. The participant sat in front of the monitor at a distance of about 54 cm, which made it possible to create a 45-degree angle between the center of the screen and the edge of the chatbot, which was located in the bottom right-hand corner of the screen. As a result, the chatbot, as assumed, was within the peripheral vision of each of the respondents. The eye tracker was calibrated so that the monitoring software would react at the right moment whenever the participant's eyesight left the designated field and returned to the correct area. This was the role of the 'overseer'. Its main purpose was to keep the user's gaze within the designated frame and not looking away into the chatbot's field.

The task of the user was to stare at the pulsating red dot in the center of the screen. They could not look outside the designated area, otherwise a warning was displayed, and until their gaze returned to the designated area, the further display of photos was suspended. When staring at the pulsating dot, the user would notice a change taking place within the chatbot in the corner of their eye, at which point they had to left-click their mouse. The information about the click time was saved by the software to a text file, which also contained accurate information about the times when individual emotions were displayed. Thus, after determining the difference between those two values, the user's reaction time to a given emotion was calculated and understood as the time between a stimulus and a response. Information was also recorded when the user looked outside the designated area and when they looked at the chatbot. There were cases when the user did not notice the changes taking place in the chatbot area—this was also recorded. The analyzed data were grouped and aggregated into three categories based on common characteristics. We included results whose transition times ranged from 100 to 2000 ms, based on a study that excluded reaction times that were <100 ms as 'too early' and reaction times that were >2000 ms as 'too late' (i.e., not a direct reaction to the stimuli) [30].

## Results

### Analysis for each emotional expression

For the analysis, we used the variables regarding changes in the chatbot field for individual time intervals, which were grouped into four variants: four repetitions, and groups of emotions (i.e., positive and negative). In Table 1, which reflects the response to changes in the chatbot field using Mann–Whitney U statistical analysis, we can see that the intergroup comparison shows a statistical significance below $p < 0.05$ in a few cases.

The intergroup comparison of the emotion of surprise with the rest of the emotions shows significance in virtually every case, except for the emotion of happiness. This indicates little variation in reaction times.

These emotions are both positive and resulted in a similar average response time. Starting the analysis at the base, we can conclude, as shown in Fig 2(A), that the average user reaction time to the change in emotions was about 0.499 s, with the reaction time to the face expressing disgust (0.526 s) and fear (0.521 s) being the longest. Users reacted the fastest to the faces expressing surprise (0.471 s) and happiness (0.472 s), while faces expressing anger and sadness resulted in reactions at a slightly slower speed (0.498 s).

**Table 1. Intergroup comparision of emotions.**

| Expression | anger | disgust | fear | happiness | surprise | sadness |
|---|---|---|---|---|---|---|
| *anger* | x | .705(.37) | .511(.65) | .278(1.08) | .066(1.83) | .891(.13) |
| *disgust* | | x | .795(.25) | .127(1.52) | .028(2.19) | .751(.31) |
| *fear* | | | x | .067(1.82) | .014(2.44) | .587(.54) |
| *happiness* | | | | x | .457(.74) | .217(1.23) |
| *surprise* | | | | | x | .051(1.95) |
| *sadness* | | | | | | x |

A comparison of all emotions. The table shows the significance ($p < 0.05$) or lack thereof of the comparison and its strength, i.e., the value, in parentheses.

Fig 2(B) shows the frequency of reaction times obtained for each of the emotions. The width of reaction time interval is 0.08 s. The legend in the upper right-hand corner indicates which color corresponds to the frequency of the recorded reactions in the given time interval. It can be seen here that the vast majority of reaction times ranged from 0.36 to 0.52 s. In this interval, most of the reactions were to the emotion of surprise. The histogram on the side shows that, above 0.5 s, the number of reactions decreases with the duration of the reaction

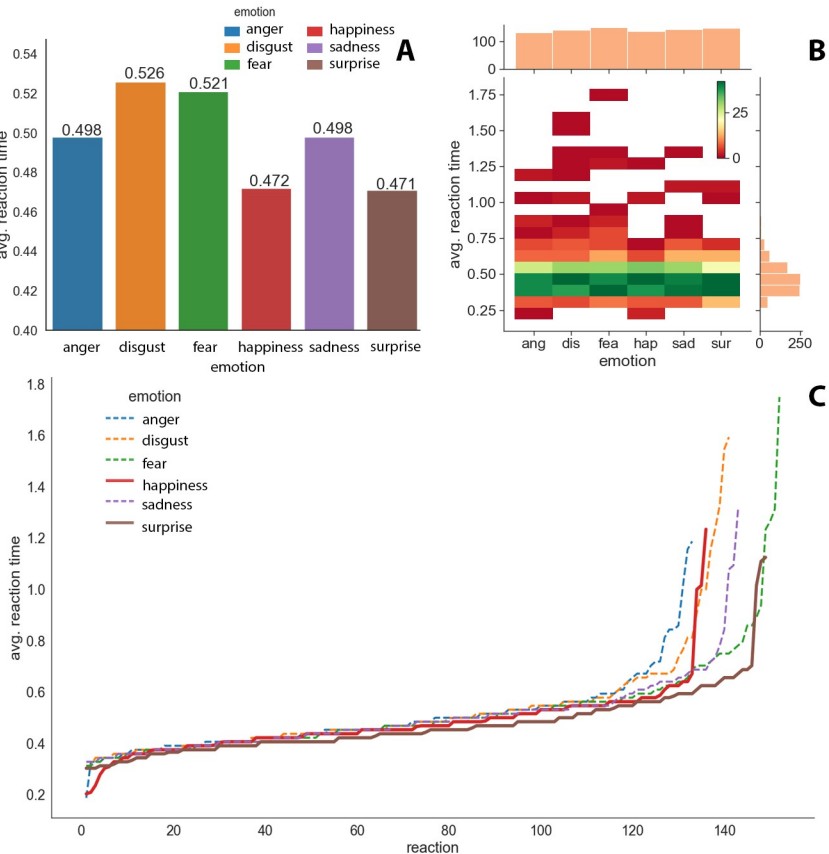

**Fig 2.** Barplot with average time reaction for each emotion (A) heatmap for each emotion (B) and line graphs (C) showing individual reaction times sorted ascending for each emotion (positive emotions presented as continuous curves and negative emotions as broken curves).

time. After about 0.7 s, the number of these reactions, regardless of the emotion, is negligible. The top histogram shows the sum of correct reactions obtained by all users. On average, there were 142.5 reactions per emotion.

Fig 2(C) shows all reaction times for each emotion. The times for each emotion were sorted from best to worst. Solid lines are used to highlight the reactions to the positive emotions of surprise and happiness. Emotions from the negative group are presented with dashed lines. One can see here that, regardless of the emotion, most of the results are similar, but a significant portion of the reactions to positive emotions achieved better results compared with the other emotions. The emotion of surprise stands out here, in particular, which is almost always below the other lines. The line of happiness is higher than that of surprise, but still stands apart from the negative emotions.

## Analysis for aggregated positive vs. negative emotions

As it was mentioned, the emotions were divided into two groups: positive and negative. Positive emotions are a group of positively associated emotions: happiness and surprise. Negative emotions, in this case, are non-positive emotions: anger, fear, disgust, and sadness. Fig 3 shows the results obtained for aggregated positive and negative emotions.

The results of the overall averages indicate that the reactions to positive emotions were much faster than those to negative emotions.

Fig 3(A) shows all reaction times for each type of aggregated emotion. The times for each type were sorted from best to worst. Solid lines represent the average results for the group. The chart also shows the area defined by the worst and best reaction outcomes from a given group. It is evident here that the group of positive emotions resulted in faster reaction times. From the areas behind the lines, it can be seen that the results for the group of positive emotions were also less varied for each of the emotions than those for the group of negative emotions, but this may be influenced by the fact that there are twice as many emotions in the negative group.

The analysis in Fig 3(B) of the intergroup comparison showed that the significance group fluctuated at the level of $p < 0.003$. This proves a clear difference between the two groups. The $Z$ factor in this case was 2.913. The average response times for the negative and positive emotion groups were 0.511 s and 0.472 s, respectively. This indicates faster responses to faces that show positive emotions.

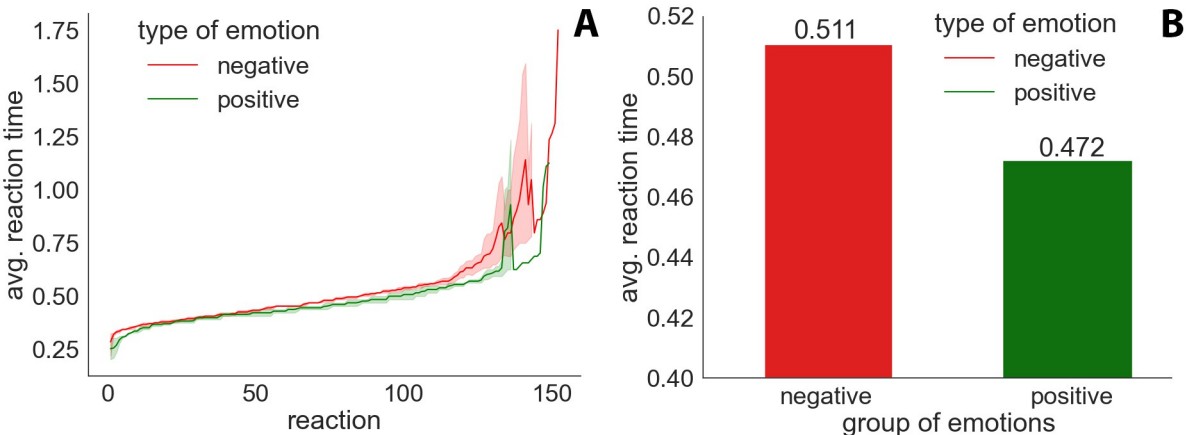

**Fig 3.** Figure (A) shows individual average reaction times (sorted ascending) for aggregated positive and negative emotions along with the coverage for individual emotions in each aggregate, and Figure (B) shows the average time for each group of emotions.

## Analysis of sequences of emotions

Fig 4(A) shows the average reaction time for each emotion in relation to the sequence number, and the heatmap in Fig 4(B) shows the frequency of reaction times obtained for each of the sequences for all emotions, with the width of the reaction time interval being 0.08 s. The legend on the right indicates what frequency of recorded reactions in a given time interval corresponds to each color.

One can infer interesting conclusions from Fig 4(A). This figure implies that for positive emotions of happiness and surprise, a decrease in the average reaction time with each repetition was observed; e.g., the reaction time in the case of happiness dropped from being close to 0.51 s to 0.45 s. For the negative emotion of fear, the reaction time decreased with each repetition from 0.55 to 0.48 s. The other negative emotions showed no similar trends. The emotion of anger at the first repetition reached a reaction time of about 0.46 s, after which this time stabilized at a level of nearly 0.5 s. In turn, for the disgust and sadness emotions, highly inconsistent results were observed.

Therefore, with regard to the emotions of happiness, surprise, and fear, users tended to 'learn' in order to elicit a quicker response. The remaining emotions, however, did not show such tendency. Rather, they were highly variable. Globally, by dividing emotions into positive and negative aggregates, we can conclude that the group of positive emotions showed a learning trend. The positive emotions had a stronger influence on the user than the negative emotions, only one of which showed a learning trend.

Fig 4(B) shows the frequency of reaction times obtained for each of the repetitions for all emotions, with the width of the reaction time interval being 0.08 s. The legend on the right indicates what frequency of recorded reactions in a given time interval corresponds to each color.

Fig 4(B) depicts the frequency of reaction times observed for all emotions in a sequence. Based on the side histogram, most reaction times oscillated in the range between 0.36 s and 0.52 s. This was most pronounced in Sequence 3. Sequence 1 seemed to be the most diverse in terms of the frequency of reaction times at different intervals. Furthermore, with each successive sequence, an increasing number of reactions were recorded mainly in the aforementioned time interval from 0.36 s to 0.52 s. There was an average of 214 reactions per sequence. It can

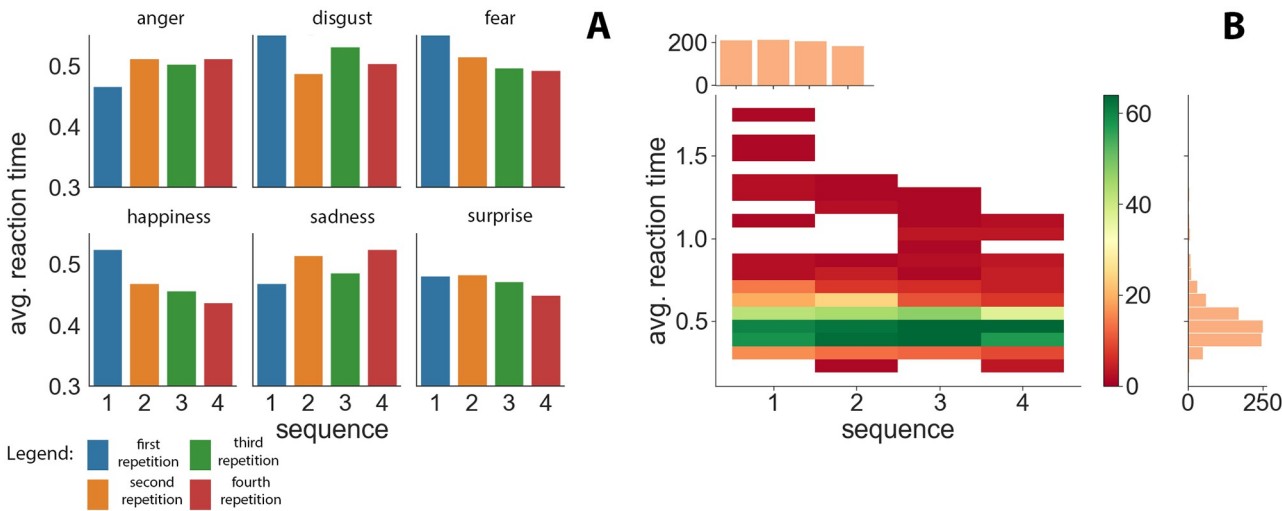

**Fig 4.** Figure (A) shows the average reaction time for each emotion in relation to the sequence number, and the heatmap (B) shows the frequency of reaction times for each of the sequences for all emotions combined.

also be seen that, in the last two sequences, the number of reactions started to decrease. In the successive sequences, 221, 223, 217, and 193 reactions were recorded sequentially.

## Discussion and conclusion

The main goal of this study was to investigate how the user's reaction time was affected by a visual change in emotions expressed through facial expressions of a chatbot represented by real human face photos located within the user's peripheral vision. Fraser W. Smith and Stephanie Rossit showed that different emotions are recognized and detected with varying results, which encouraged our studies [29]. Our results show that faces expressing happiness and surprise were noticed the fastest. Their results convincingly demonstrated that task constraints shape perception of expression in the peripheral vision and provided new evidence that detection and recognition relies on separate underlying mechanisms, with recognition being more stimulus-dependent [29]. Bayle et al. showed that humans have the ability to detect the presence and type of emotion according to the facial expressions of a presented photo in the far periphery [28]. Much research has focused on affect rather than how humanoid objects are perceived within chatbots [11, 31, 32].

Both of these emotions of happiness and surprise can be classified as positive, while the others, which were also present in the study, i.e., anger, disgust, sadness, and fear, are classified as negative or pejorative. By analyzing the emotions in groups, the study showed that changes to emotions in the positive group were noticed 7.6% faster than those in the negative group. As the emotion changes were displayed to users in sequences, the study revealed that the emotions of happiness, surprise, and fear with each successive change led to a decrease in the average response time of users. No tendency was noted for the remaining emotions. Without classifying the emotions, the average response time to visual changes in a given sequence still decreased with each successive change. The average reaction time for the last sequences compared to the first ones was 6.2% faster. The reason for this may be an effect of the user learning/recognizing a given emotion. This study suggests that, when developing a humanoid chatbot model that can express certain emotions, it is worth considering which emotions should be selected for specific messages if a specific and prompt reaction in the user is desired.

We have shown that some expressions are detected faster in the peripheral vision, i.e., happiness and surprise, and some are detected more slowly, i.e., anger, sadness, disgust, and fear, the last two showing the longest reaction times. Our work examined the performance of activities occurring within the user's peripheral vision.

The researched area is represented in a small number of studies. This paper is an extension of the research to date, and is particularly related to Bayle's work on changes in emotional states [28]. This follow-up study was based on adding humanoid features to the chatbot area with the main focus on emotional changes, which proved to be an effective way to catch the attention of the user in another area [29].

An important aspect of chatbots is to increase the naturalness of their assistance and communication with the user. McCrickard et al. [25] discussed utility as a value provided by the peripheral system as a whole and did not directly manipulate utility as part of their experiment, whereas we considered utility as the meaning of the content of the individual gaps; therefore, the concept of utility should be better defined. The emotions that are shown in the chatbot area clearly influence the user's reaction speed. It was found that particular emotions seem to stimulate or slow down the interactions located in the peripheral vision.

In terms of guidelines for interface designers, recommender system programmers, and content creators of chatbots based on research, they should consider making the images of people representing the chatbot positive. This will affect reaction speed.

Game designers may also create guidelines for their creations. Messages appearing with adequate avatars that express different emotions in the peripheral vision may affect the speed of the player's reaction. In the excitement of gameplay in MMO (massively multiplayer online) games or even simulators, e.g., piloting a plane or driving a vehicle, every fraction of a second counts. In MMO games there are often various messages with different icons and the gamer needs to perceive peripheral changes to be able to react quickly, e.g., by pressing key combinations or by clicking the appropriate icon, or in some games, e.g., StarCraft, the character's avatar at the bottom of the screen suggests their 'damage', which can affect the decisions made by the user. One way to combine theories about avatars in chatbots and simulators of simulation tasks in relation to the peripheral vision is to use the avatar's peripheral vision as a way to display information about the simulation task.

For example, in a virtual reality driving simulation, the user's avatar could have a heads-up display that shows their speed and other important information, but it could also use the avatar's peripheral vision to display information about the environment, such as traffic signals or other vehicles in the vicinity. This could help the user to more easily stay aware of their surroundings and make more informed decisions while driving. Additionally, the chatbot feature of the avatar could be used to provide verbal cues or warnings, making use of the peripheral vision in a more natural way. By introducing appropriate content expressing individual emotions, we can aim to extend the reaction time or effectively shorten it. However, users should not be overloaded with other stimuli, e.g., smiles and humor, within the chatbot area, because repeated and too invasive representations of emotions often result in habituation [33–35].

In the presented study, we aimed to analyze the performance of a peripherally located chatbot, represented by human face photos, in terms of the impact of changes in the chatbot's emotional expressions on the user's reaction. The goal in itself was to study the user's reaction to changes in the chatbot field; however, users had no idea that the changing photos in the peripheral vision represented different emotions. Our earlier study was based on the impact of changes in textual-only content within the chatbot, and we view this study as an extension to that research. The results showed that users are more sensitive to positive emotions within chatbots, as they are noticed more quickly than negative facial expressions.

The presented study raises questions for further research related to the impact of techniques used for avatar visualization, e.g., simplified forms vs. realistic avatars, on user reactions. Other research could also focus on another form of emotion expression, e.g., avatar gestures or symbols conveying emotional messages.

## Author Contributions

**Conceptualization:** Kacper Fornalczyk, Jarosław Jankowski.

**Data curation:** Kamil Bortko, Kacper Fornalczyk, Jarosław Jankowski, Karina Dziedziak.

**Formal analysis:** Kamil Bortko, Kacper Fornalczyk, Jarosław Jankowski, Karina Dziedziak.

**Methodology:** Jarosław Jankowski, Piotr Sulikowski.

**Visualization:** Kamil Bortko, Kacper Fornalczyk, Jarosław Jankowski.

**Writing – original draft:** Kamil Bortko, Kacper Fornalczyk, Jarosław Jankowski.

**Writing – review & editing:** Kamil Bortko, Kacper Fornalczyk, Jarosław Jankowski, Piotr Sulikowski.

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
