## [Decision Letter · Decision Letter 0]

15 Dec 2022

PONE-D-22-20948Impact of chatbot facial emotions changes on user attention and reaction timePLOS ONE

Dear Dr. Bortko,

Thank you for submitting your manuscript to PLOS ONE. After careful consideration, we feel that it has merit but does not fully meet PLOS ONE’s publication criteria as it currently stands. Therefore, we invite you to submit a revised version of the manuscript that addresses the points raised during the review process.

 Two Reviewers evaluated the manuscript. I encourage Authors to provide a revised version, paying particular attention to clarity in methodology and the data availability issue. 

We look forward to receiving your revised manuscript.

Kind regards,

Stefano Triberti, Ph.D.

Academic Editor

PLOS ONE

Journal Requirements:

2. Please note that PLOS ONE has specific guidelines on code sharing for submissions in which author-generated code underpins the findings in the manuscript. In these cases, all author-generated code must be made available without restrictions upon publication of the work. Please review our guidelines at https://journals.plos.org/plosone/s/materials-and-software-sharing#loc-sharing-code and ensure that your code is shared in a way that follows best practice and facilitates reproducibility and reuse. New software must comply with the Open Source Definition.

4. Please ensure that you have specified (1) whether consent was informed and (2) what type you obtained (for instance, written or verbal, and if verbal, how it was documented and witnessed). If your study included minors, state whether you obtained consent from parents or guardians. If the need for consent was waived by the ethics committee, please include this information.

This work was supported by the National Science Centre of Poland, Decision No.

2017/27/B/HS4/01216 ( J.J., K.B.) and this research was supported by ZUT Highfliers

School /Szko la Or l´ow ZUT/ project co-ordinated by Dr. Piotr Sulikowski, within the

framework of the program of the Minister of Education and Science /Grant No.

MNiSW/2019/391/DIR/KH, POWR.03.01.00-00-P015/18/, co-financed by the

European Social Fund, the amount of financing PLN 1,704,201,66 (K.F.).

However, funding information should not appear in the Acknowledgments section or other areas of your manuscript. We will only publish funding information present in the Funding Statement section of the online submission form. 

This work was supported by the National Science Centre of Poland, Decision No. 2017/27/B/HS4/01216 ( J.J., K.B.) and this research was supported by ZUT Highfliers School /Szkoła Orłów ZUT/ project co-ordinated by Dr. Piotr Sulikowski, within the framework of the program of the Minister of Education and Science /Grant No. MNiSW/2019/391/DIR/KH, POWR.03.01.00-00-P015/18/, co-financed by the European Social Fund, the amount of financing PLN 1,704,201,66 (K.F.).

Reviewers' comments:

Reviewer's Responses to Questions

**Comments to the Author**

1. Is the manuscript technically sound, and do the data support the conclusions?

Reviewer #1: Partly

Reviewer #2: Yes

2. Has the statistical analysis been performed appropriately and rigorously? 

Reviewer #1: N/A

Reviewer #2: Yes

3. Have the authors made all data underlying the findings in their manuscript fully available?

Reviewer #1: Yes

Reviewer #2: No

4. Is the manuscript presented in an intelligible fashion and written in standard English?

Reviewer #1: No

Reviewer #2: Yes

5. Review Comments to the Author

Reviewer #1: This paper analyses the impact of emotional changes in a chatbot representation on users’ attention and reaction times. The study deals with a very interesting and timely topic. The literature on the impact of conversational agents is rapidly expanding in different areas, and the question of how their visual appearance affects user experience is crucial. However, I have some concerns about the research execution, the paper’s structure, and the interpretation of the findings.

1. In the abstract and the introduction sections, the paper appears to focus on how specific visual characteristics (in this case, changes in emotional expressions) of a chatbot attract users’ attention and make the chatbot get noticed among many other possible competing visual information displayed on a website or an online platform. The authors indeed state: “While chatbots compete with other graphic elements within websites or applications, attracting user attention is a challenge […]. Drawing attention to a chatbot area localised within the periphery area can be based on the usage of different visual characteristics […] Expressed emotions can be another technique to attract user attention […]”. However, results are mainly discussed in terms of perceptual processes, such as emotion detection and recognition in the periphery of the visual system. Both in the introduction and discussion sections, the authors should better clarify the link between emotional change detection and users’ attention. If the main goal was to understand which visual characteristics of a chatbot make it more noticeable and direct users’ attention towards it, how should reactions times to changes in the chatbot’s emotional expressions measure this? It is likely that participants are aware of the presence of the chatbot regardless of the emotion displayed but that they are only detecting a change to positive emotion faster than negative emotions.

2. It is also unclear whether the authors refer to visual characteristics in terms of “emotional change” or “expressed emotional states”. As mentioned in the text, the goal of the study was to verify “the impact of emotional changes within a chatbot’s face on user attention and reaction time”. The experiment setup seems to be heading in this direction. Participants were indeed instructed to click the mouse when they perceived a “change” in the chatbot’s emotional expression. Results are also partially discussed in terms of how fast participants were in detecting this change (e.g., “As the emotion changes were displayed to users in sequences, the study revealed that the emotions of happiness, surprise and fear with each successive change led to a decrease in the average response time of users”). On the other hand, though, the authors discuss their results in terms of expressed emotional state (e.g., “When developing a humanoid chatbot model that can express certain emotions, it is worth considering which emotions should be selected for specific messages if a specific reaction in the user is the desired”). In this case, the authors discuss the results in terms of which type of emotional state displayed by a chatbot is more effective in eliciting a specific reaction (authors should also clarify to the reader what they mean by “specific reaction”: reaction times? Emotional and/or social response?). Therefore, it is unclear whether the focus is on the effect of users’ detection of changes in emotions or on the impact of the emotional state displayed by the chatbot. The authors should clarify this point.

3. I find it challenging to understand the transition from talking about chatbots used in online platforms and websites for communication purposes to the guidelines for game designers outlined in the conclusion section. First, if the authors intend to generalise their results to the context of video games, they should explain the role of avatars displayed in the periphery. Second, they state: “In the excitement of gameplay in MMO games or even simulators, e.g., piloting a plane or driving a vehicle, every fraction of second counts. By introducing appropriate content expressing individual emotions, we can extend the reaction or effectively shorten it”. This conclusion was quite confusing to me. Specifically, it is unclear whether the outcome variables (user attention and reaction times) are investigated in relation to the central task or the chatbot (drawing the user’s attention towards it). Why should a peripheral avatar displaying certain emotions (or emotional changes) affect the performance on the simulation task (e.g., driving or piloting) in this case?

4. The authors leave out important information regarding the chatbot’s design. There needs to be much more discussion on the chatbot’s appearance. In Figure 1, the chatbot appears as a gender-neutral emoticon, though in the text, it is specified that they used male and female faces (Male 1, Male 2, Female 1, Female 2). In the introduction, the authors refer to anthropomorphism and the uncanny valley theory, and further refer to the chatbot visualisation in terms of “graphic avatar” and “humanoid chatbot”, which makes me assume that they used computer-generated human-like faces. Details on the chatbots’ appearance should include the software and technique used and information on realism (e.g., in terms of texture) with which they were created. The degree of realism of a human-like digital entity generates very different reactions in the observers and it has different implications in terms of the uncanny valley effect. The authors should clarify this section to avoid confusion.

5. In the introduction section, the authors refer the potential of chatbots’ human-like features to help create a more authentic and natural interaction and communication. The authors may consider the “computers as social actors” and “social presence” theory as possible theoretical framework to explain why embedding social cues (e.g., human-like avatar showing emotional expressions) in chatbots’ design enhances users’ engagement and promotes a more natural interaction.

6. I found the manuscript poorly organised. Many sentences disrupt the flow of the text and raise several questions. A couple of examples:

- “The visualisation of chatbots and their social features, should target users’ expectations, ultimately avoiding frustration and dissatisfaction”. Expectation to what? Also, why are the authors here referring to frustration and dissatisfaction? I would expect a link between the focus of the study (users’ attention) and these outcomes.

- “Additionally, there is positive relationship between human likeness and knowledge”. What does this mean and how is it related to what the authors are saying?

7. The language is unclear, with many imprecisions and hard-to-read or incomplete sentences, making it difficult to follow. I would recommend language editing.

8. Overall, I think the authors are addressing a very interesting topic, but need to restructure the introduction and discussion and clarify some main points before drawing any conclusion. The theories and related findings are indeed sparse and therefore do not support the research framework of this study. Hypotheses development is also not substantiated. Due to these critical issues, the discussion fails to show relevance to the literature as the links between predictions and findings with related theories and findings from similar works are missing.

Reviewer #2: This article presents a study that evaluates the differences in time reaction in the detection of a chatbot (presented in a periphery area of the screen) if it shows different emotions.

The authors provide detailed motivations and references, which led to the experiment's construction, in a precise and logical way. The methodology and results appear sound.

However, as per “e PLOS Data policy” “all data underlying the findings described in the manuscript need to be fully available”, but I was not able to find them, either reported in the article or through an indication of where they could be found (e.g. public repository); also the Data Availability Statement does not describe where the data can be found. If specific reasons make the data not disclosable, they should be described accurately. If not, data or an indication of where they can be consulted should be included.

Furthermore, there are some minor issues I think need to be addressed before publication:

1) the sample should be described with more details (es. gender, inclusion and exclusion criteria)

2) some captions and figures are unclear and are missing measurement units (e.g. ms):

2a) fig 2 B the color code of the chart and the legend of the two histograms on top and on the right should be included

2b) fig 2 C does not look like a “scatterplot”, I suggest an alternative definition. Furthermore, a better definition of “reaction” is needed and the caption is unclear

2c) fig 3 A in the caption “shows two groups of emotions (positive and negative) along with the coverage of individual groups of emotions” should be more explicit; a description of the light green and light red should be provided also in a legend

2d) fig 3 B the color code of the chart and the legend of the two histograms on top and on the right should be included

3) In the following phrases

3a) “In fact, Mori observed an increased sensitivity to defects in a human-like form”: which type of defects are they referring to?

3b) “Additionally, there is a positive relationship between human likeness and knowledge”: the meaning is obscure

3c) “In detection, however, while happiness and surprise are still well detected, fear is also a well detected expression”: there seems to be an error here

3d) “Both of these emotions can be classified as positive, while the others, which were also present in the study, i.e., anger, disgust, sadness, and fear, are negative or pejorative”: the phrase does not link with the one before

3e) An important aspect is to increase the naturalness of the chatbot’s communication with the user: the phrase does not link with the one before

3f) “McCrickard et al. discussed utility as a value provided by the peripheral system as a whole and did not directly manipulate utility as part of their experiment, whereas we consider utility as the meaning of the content of the individual gaps”: the concept of utility should be better defined

4) Proofreading is suggested. For example, the form of the following phrases should be reviewed:

4a) “The graphic avatar in the chatbot field will have a different impact on a conversation than a conversation with a human and increases the fluency of conversation”

4b) “This secondary task is usually a given, with the task supported by the peripheral system being evaluated”

4c) “In real systems, the chatbot, as well as system messages or in-video-game messages, aimed at helping users or players, is targeted at a more accurate peripheral search and identification tasks while performing a central search”

4d) “Starting the analysis at the base, we can conclude in Figure 2 (A)“

4e) ”It can be seen here that the vast majority of reaction times ranged from 0.36 to 0.52 s. In this interval, most reactions were to the emotion of surprise”

6. PLOS authors have the option to publish the peer review history of their article (what does this mean?). If published, this will include your full peer review and any attached files.

Reviewer #1: No

Reviewer #2: No

---

## [Author Response · Author response to Decision Letter 0]

7 Mar 2023

To editor:

1. We note that several of your files are duplicated on your submission. Please remove any unnecessary or old files from your revision, and make sure that only those relevant to the current version of the manuscript are included.

We also notice that your manuscript file was uploaded on July 25, 2022. Please can you upload the latest version of your revised manuscript as the main article file, ensuring that does not contain any tracked changes or highlighting. This will be used in the production process if your manuscript is accepted.

Please follow this link for more information: http://blogs.PLOS.org/everyone/2011/05/10/how-to-submit-your-revised-manuscript/

Answer:

I have added the current files to the review.

2. Please note that PLOS ONE has specific guidelines on code sharing for submissions in which author-generated code underpins the findings in the manuscript. In these cases, all author-generated code must be made available without restrictions upon publication of the work. Please review our guidelines at https://journals.plos.org/plosone/s/materials-and-software-sharing#loc-sharing-code and ensure that your code is shared in a way that follows best practice and facilitates reproducibility and reuse. New software must comply with the Open Source Definition.

Answer: 

We performed the test using an eyetracker. Data is attached in https://github.com/kbzut/Periperial-chatbot-face

Answer:

The study was conducted according to the guidelines of the Declaration of Helsinki and approved by the Ethics Committee of Bioethical Commission of Pomeranian Medical University, Szczecin (09.03.2020); Bioethics Committee Agreement No. KB-0012/24/2020.

4. Please ensure that you have specified (1) whether consent was informed and (2) what type you obtained (for instance, written or verbal, and if verbal, how it was documented and witnessed). If your study included minors, state whether you obtained consent from parents or guardians. If the need for consent was waived by the ethics committee, please include this information.

Answer:

 Informed consent was obtained from all subjects involved in the study. Written informed consent was obtained from the patient(s) to publish this paper. The research described for the papers was accepted by Bioethics Committee: agreement No.~KB-0012/24/2020.

We did not have minors in our study.

This work was supported by the National Science Centre of Poland, Decision No.

2017/27/B/HS4/01216 ( J.J., K.B.) and this research was supported by ZUT Highfliers

School /Szko la Or l´ow ZUT/ project co-ordinated by Dr. Piotr Sulikowski, within the

framework of the program of the Minister of Education and Science /Grant No.

MNiSW/2019/391/DIR/KH, POWR.03.01.00-00-P015/18/, co-financed by the

European Social Fund, the amount of financing PLN 1,704,201,66 (K.F.).

However, funding information should not appear in the Acknowledgments section or other areas of your manuscript. We will only publish funding information present in the Funding Statement section of the online submission form.

This work was supported by the National Science Centre of Poland, Decision No. 2017/27/B/HS4/01216 ( J.J., K.B.) and this research was supported by ZUT Highfliers School /Szkoła Orłów ZUT/ project co-ordinated by Dr. Piotr Sulikowski, within the framework of the program of the Minister of Education and Science /Grant No. MNiSW/2019/391/DIR/KH, POWR.03.01.00-00-P015/18/, co-financed by the European Social Fund, the amount of financing PLN 1,704,201,66 (K.F.).

Answer:

We removed this information form our manuscript.

Answer:

Data is attached in https://github.com/kbzut/Periperial-chatbot-faces

Response to Reviewer 1 Comments

At the beginning, we would like to thank the Reviewer for valuable comments

and tips. We appreciate the time spent and helpful remarks. Please find the

answers and comments to the doubts presented in the review enclosed below.

Point 1: In the abstract and the introduction sections, the paper appears to

focus on how specific visual characteristics (in this case, changes in emotional

expressions) of a chatbot attract users’ attention and make the chatbot get

noticed among many other possible competing visual information displayed on

a website or an online platform. The authors indeed state: “While chatbots

compete with other graphic elements within websites or applications, attracting

user attention is a challenge [. . . ]. Drawing attention to a chatbot area localised

within the periphery area can be based on the usage of different visual char-

acteristics [. . . ] Expressed emotions can be another technique to attract user

attention [. . . ]”. However, results are mainly discussed in terms of perceptual

processes, such as emotion detection and recognition in the periphery of the

visual system. Both in the introduction and discussion sections, the authors

should better clarify the link between emotional change detection and users’

attention. If the main goal was to understand which visual characteristics of

a chatbot make it more noticeable and direct users’ attention towards it, how

should reactions times to changes in the chatbot’s emotional expressions mea-

sure this? It is likely that participants are aware of the presence of the chatbot

regardless of the emotion displayed but that they are only detecting a change

to positive emotion faster than negative emotions.

Response 1: We would like to clarify that users participating in the study focused

their eyes on the left side of the screen. In the periphery, they noticed changing

pictures with expressed emotions in the chatbot window. They could presume

that there is a chatbot there, however, the experiment was set up so that they

could not look directly at the chatbot because of validation and real time control

of xy position of gaze patterns. The aim of the study was to notice which

changes within a chatbot area make it more noticeable while examining the

response times to changes in the chatbot’s emotional expression.

Point 2: It is also unclear whether the authors refer to visual characteristics in

terms of “emotional change” or “expressed emotional states”. As mentioned in

the text, the goal of the study was to verify “the impact of emotional changes

within a chatbot’s face on user attention and reaction time”. The experiment

setup seems to be heading in this direction. Participants were indeed instructed

to click the mouse when they perceived a “change” in the chatbot’s emotional

expression. Results are also partially discussed in terms of how fast participants

were in detecting this change (e.g., “As the emotion changes were displayed to

users in sequences, the study revealed that the emotions of happiness, surprise

and fear with each successive change led to a decrease in the average response

time of users”). On the other hand, though, the authors discuss their results in

terms of expressed emotional state (e.g., “When developing a humanoid chatbot

model that can express certain emotions, it is worth considering which emotions

should be selected for specific messages if a specific reaction in the user is the

desired”). In this case, the authors discuss the results in terms of which type

of emotional state displayed by a chatbot is more effective in eliciting a specific

reaction (authors should also clarify to the reader what they mean by “specific

reaction”: reaction times? Emotional and/or social response?). Therefore, it

is unclear whether the focus is on the effect of users’ detection of changes in

emotions or on the impact of the emotional state displayed by the chatbot. The

authors should clarify this point.

Response 2: The presented study analyzed the impact of changes in emotional

expression in the chatbot representation and their impact on the user’s reaction

to these changes that the user saw in the peripheral area. The goal in itself

was to study the reaction to changes in the chatbot field, however, users had

no idea that the changing pictures represented different emotions. Our earlier

study was based on the impact of changes of textual content within the chatbot

and we treat this paper as a continuation of that research direction, as it is now

explained in the paper.

Point 3: I find it challenging to understand the transition from talking about

chatbots used in online platforms and websites for communication purposes to

the guidelines for game designers outlined in the conclusion section. First, if the

authors intend to generalise their results to the context of video games, they

should explain the role of avatars displayed in the periphery. Second, they state:

“In the excitement of gameplay in MMO games or even simulators, e.g., piloting

a plane or driving a vehicle, every fraction of second counts. By introducing

appropriate content expressing individual emotions, we can extend the reaction

or effectively shorten it”. This conclusion was quite confusing to me. Specifi-

cally, it is unclear whether the outcome variables (user attention and reaction

times) are investigated in relation to the central task or the chatbot (drawing

the user’s attention towards it). Why should a peripheral avatar displaying cer-

tain emotions (or emotional changes) affect the performance on the simulation

task (e.g., driving or piloting) in this case?

Response 3: Here we wanted to refer directly to visual messages in various

forms - often in MMO games there are messages with different icons, e.g. , the

gamer perceives peripheral changes and is able to react quickly, e.g. by pressing

key combinations or by clicking on the appropriate icon. Or in a game, for

example, Starcraft - the character’s avatar at the bottom of the screen suggests

their ”damage”, which can affect the decisions made by the user. One way to

combine theories about avatars in chatbots and simulators of simulation tasks

in relation to peripheral vision is to use the avatar’s peripheral vision as a way

to display information about the simulation task. For example, in a virtual

reality driving simulation, the user’s avatar could have a heads-up display that

shows their speed and other important information, but it could also use the

avatar’s peripheral vision to display information about the environment, such as

traffic signals or other vehicles in the vicinity. This could help the user to more

easily stay aware of their surroundings and make more informed decisions while

driving. Additionally, the chatbot functionality of the avatar could be used to

provide verbal cues or warnings, making use of the peripheral vision in a more

natural way.

Point 4: The authors leave out important information regarding the chatbot’s

design. There needs to be much more discussion on the chatbot’s appearance. In

Figure 1, the chatbot appears as a gender-neutral emoticon, though in the text,

it is specified that they used male and female faces (Male 1, Male 2, Female 1,

Female 2). In the introduction, the authors refer to anthropomorphism and the

uncanny valley theory, and further refer to the chatbot visualisation in terms of

“graphic avatar” and “humanoid chatbot”, which makes me assume that they

used computer-generated human-like faces. Details on the chatbots’ appearance

should include the software and technique used and information on realism (e.g.,

in terms of texture) with which they were created. The degree of realism of a

human-like digital entity generates very different reactions in the observers and

it has different implications in terms of the uncanny valley effect. The authors

should clarify this section to avoid confusion.

Response 4: We should explain that we used real photos of people. The emoji

is used to suggest an emotional state, not a computer-generated avatar. We

had pictures of several people in different emotional states however we did not

include real photos in the paper due to privacy concerns of the subjects. Please

see the images here https://github.com/kbzut/Periperial-chatbot-faces.

Point 5: In the introduction section, the authors refer the potential of chatbots’

human-like features to help create a more authentic and natural interaction and

communication. The authors may consider the “computers as social actors”

and “social presence” theory as possible theoretical framework to explain why

embedding social cues (e.g., human-like avatar showing emotional expressions)

in chatbots’ design enhances users’ engagement and promotes a more natural

interaction.

Response 5: Yes, the ”computers as social actors” theory and ”social presence”

theory are possible theoretical frameworks that could be used to explain why

embedding social cues, such as human-like avatars showing emotional expres-

sions, in chatbots’ design enhances users’ engagement and promotes a more

natural interaction. The ”computers as social actors” theory posits that people

tend to attribute human-like characteristics, such as intentions and emotions, to

computers and other technology. This can lead to a more engaging and natural

interaction with a chatbot, as users may feel like they are communicating with

a human-like entity. The ”social presence” theory suggests that the degree of

social presence in a communication medium can affect the perceived level of

social presence in the interaction. Chatbots with human-like avatars and so-

cial cues may increase the perceived social presence, making the interaction feel

more authentic and natural. Both theories suggest that by making chatbots

appear more human-like, users may be more likely to engage with them and

have a more authentic and natural interaction. These theories can be a useful

framework for understanding why chatbots with social cues, such as human-like

avatars, are more engaging and effective in promoting natural interaction. We

suggested publications such as ”Design of chatbot with 3D avatar, voice inter-

face, and facial expression” and ”Dynamic human and avatar facial expressions

elicit differential brain responses” to assert that the presence of a human avatar

in chatbots inspires greater user confidence.

Point 6: I found the manuscript poorly organised. Many sentences disrupt the

flow of the text and raise several questions. A couple of examples: - “The

visualisation of chatbots and their social features, should target users’ expec-

tations, ultimately avoiding frustration and dissatisfaction”. Expectation to

what? Also, why are the authors here referring to frustration and dissatisfac-

tion? I would expect a link between the focus of the study (users’ attention) and

these outcomes. - “Additionally, there is positive relationship between human

likeness and knowledge”. What does this mean and how is it related to what

the authors are saying?

Response 6: We improved organisation and mistakes in our language which

caused the initial confusion. The effect hypothesis is that humanoid objects

that look almost, but not quite, like real people make observers experience eerie

or strangely familiar feelings of revulsion or eerieness. In fact, Mori observed a

degree of defect-proneness in near-human forms - an ”uncanny valley” in what

otherwise follows from the relationship between anticipation and familiarity.

Since then, research has focused on affect, but still little is known about how

humanoid objects are actually perceived, in particular in chatbots.

Point 7: The language is unclear, with many imprecisions and hard-to-read or

incomplete sentences, making it difficult to follow. I would recommend language

editin.

Response 7: We have made a linguistic correction.

Point 8: Overall, I think the authors are addressing a very interesting topic,

but need to restructure the introduction and discussion and clarify some main

points before drawing any conclusion. The theories and related findings are

indeed sparse and therefore do not support the research framework of this study.

Hypotheses development is also not substantiated. Due to these critical issues,

the discussion fails to show relevance to the literature as the links between

predictions and findings with related theories and findings from similar works

are missing.

Response 8: The studies area is represented in a low number of studies. We treat

our paper as a kind of follow up related to paper with changes to emotional states

presented in the paper ”Emotional facial expression detection in the peripheral

visual field”. Our extension is based on adding humanoid features to the chatbot

area with main focus on emotional states changes which proved to be an effective

way to catch attention in another area.

Response to Reviewer 2 Comments

At the beginning, we would like to thank the Reviewer for valuable comments

and tips. We appreciate the time spent and helpful remarks. Please find the

answers and comments to the doubts presented in the review enclosed below.

Point 1: the sample should be described with more details (es. gender, inclusion

and exclusion criteria)

Response 1: Regarding gender, 50% of study participants were female, and 50%

were male. Inclusion criteria could be defined as ”participants with normal or

corrected-to-normal vision and between 18 to 35 years old”. Exclusion criteria

could be defined as ”participants with a history of eye disease or neurological

disorder, or any other condition that could affect the results of the study”.

Point 2: some captions and figures are unclear and are missing measurement

units (e.g. ms)

Response 2: We have corrected that throughout the article.

Point 2a:fig 2 B the color code of the chart and the legend of the two histograms

on top and on the right should be included

Response 2a: We have corrected this.

Point 2b:fig 2 C does not look like a “scatterplot”, I suggest an alternative

definition. Furthermore, a better definition of “reaction” is needed and the

caption is unclear.

Response 2b:We have corrected this. The reaction is pressing the mouse button

when there was a change of the object (in our task, the photo with the expressed

emotion) in the chatbot that appeared in the peripheral area.

Point 2c: fig 3 A in the caption “shows two groups of emotions (positive and

negative) along with the coverage of individual groups of emotions” should be

1

more explicit; a description of the light green and light red should be provided

also in a legend

Response 2c: We have corrected this. Positive emotions are a group of posi-

tively associated emotions. Negative emotions, in this case, are ”non-positive”

emotions. Hence, they are in opposition to one another.

Point 2d: fig 3 B the color code of the chart and the legend of the two histograms

on top and on the right should be included

Response 2d: We have corrected this.

Point 3a: “In fact, Mori observed an increased sensitivity to defects in a human-

like form”: which type of defects are they referring to?

Response 3a: Mori observed that the more humanlike an object is, the more

more familiar it seems, but only until a certain point at which defects understood

as subtle deviations from human norms cause them to look disturbing.

Point 3b: “Additionally, there is a positive relationship between human likeness

and knowledge”: the meaning is obscure

Response 3b: The effect hypothesis is that humanoid objects that look almost,

but not quite, like real people make observers experience eerie or strangely

familiar sensations of revulsion or eerieness. In fact, Mori observed a degree

of vulnerability to defects in near-human forms - an ”uncanny valley” in what

otherwise follows the relationship between prediction and familiarity. Since

then, research has focused on affect, but still little is known about how humanoid

objects are actually perceived.

Point 3c: “In detection, however, while happiness and surprise are still well

detected, fear is also a well detected expression”: there seems to be an error

here

Response 3c: We have corrected the text.

Point 3d: “Both of these emotions can be classified as positive, while the others,

which were also present in the study, i.e., anger, disgust, sadness, and fear, are

negative or pejorative”: the phrase does not link with the one before

Response 3d: We have corrected the text.

Point 3e: An important aspect is to increase the naturalness of the chatbot’s

communication with the user: the phrase does not link with the one before

Response 3e: We have corrected the text.

Point 3f: “McCrickard et al. discussed utility as a value provided by the pe-

ripheral system as a whole and did not directly manipulate utility as part of

their experiment, whereas we consider utility as the meaning of the content of

the individual gaps”: the concept of utility should be better defined

Response 3f: We have developed that in the paper. In studying peripheral or

secondary display information representations, many researchers focus only on

information gained, without measuring the changes in primary task performance

caused by these displays (Gray and Saltzman, 1998). Similarly, some studies

create an unrealistic testing environment given that the distinction between

reaction and comprehension is unclear. While reaction and comprehension tasks

are often closely related, the two objectives may imply differences to notification

system information design.

Point 4a: The graphic avatar in the chatbot field will have a different impact

on a conversation than a conversation with a human and increases the fluency

of conversation”

Response 4a: We have corrected the text.

Point 4b: “This secondary task is usually a given, with the task supported by

the peripheral system being evaluated”

Response 4b:We have corrected the text.

Point 4c: “In real systems, the chatbot, as well as system messages or in-video-

game messages, aimed at helping users or players, is targeted at a more accurate

peripheral search and identification tasks while performing a central search”

Response 4c:We have corrected the text.

Point 4d: “Starting the analysis at the base, we can conclude in Figure 2 (A)“

Response 4d:We have corrected the text.

Point 4e: ”It can be seen here that the vast majority of reaction times ranged

from 0.36 to 0.52 s. In this interval, most reactions were to the emotion of

surprise”

Response 4e:We have corrected the text.

---

## [Decision Letter · Decision Letter 1]

28 Apr 2023

PONE-D-22-20948R1Impact of changes in chatbot's facial expressions on user attention and reaction timePLOS ONE

Dear Dr. Bortko,

Thank you for submitting your manuscript to PLOS ONE. After careful consideration, we feel that it has merit but does not fully meet PLOS ONE’s publication criteria as it currently stands. Therefore, we invite you to submit a revised version of the manuscript that addresses the points raised during the review process.

One Reviewer has suggested minor revision to the manuscript. I invite Authors to provide modifications in order to proceed. 

We look forward to receiving your revised manuscript.

Kind regards,

Stefano Triberti, Ph.D.

Academic Editor

PLOS ONE

Journal Requirements:

Reviewers' comments:

Reviewer's Responses to Questions

**Comments to the Author**

1. If the authors have adequately addressed your comments raised in a previous round of review and you feel that this manuscript is now acceptable for publication, you may indicate that here to bypass the “Comments to the Author” section, enter your conflict of interest statement in the “Confidential to Editor” section, and submit your "Accept" recommendation.

Reviewer #1: (No Response)

Reviewer #2: All comments have been addressed

2. Is the manuscript technically sound, and do the data support the conclusions?

Reviewer #1: Yes

Reviewer #2: Yes

3. Has the statistical analysis been performed appropriately and rigorously? 

Reviewer #1: N/A

Reviewer #2: Yes

4. Have the authors made all data underlying the findings in their manuscript fully available?

Reviewer #1: Yes

Reviewer #2: Yes

5. Is the manuscript presented in an intelligible fashion and written in standard English?

Reviewer #1: Yes

Reviewer #2: Yes

6. Review Comments to the Author

Reviewer #1: I would like to thank the authors for their effort in improving the manuscript and for clarifying some main points. However, there are still some issues that the authors should address:

At p. 2/13 “The ’social presence’ theory [11] suggests that the degree of social presence in a communication medium can affect the perceived level of social presence in the interaction”. This sentence is unclear as the authors are saying that “the degree of social presence […] can affect the level of social presence”, which is a tautology. The social presence theory suggests that the degree to which a medium (such as a chatbot) is designed to look and act more like a human, with features such as a human-like appearance, facial expressions, and gestures, positively impact the level of social presence perceived in the interaction. As a result, users may feel like they are engaging in a more natural and authentic conversation, as if they were talking to a real human instead of a machine.

At p. 2/13 “In fact, Mori [12] observed a degree of defect proneness in near-human forms an ”uncanny valley” in what otherwise follows from the relationship between anticipation and familiarity”. That is not entirely precise. The Uncanny Valley (UV) hypothesis refers to the relationship between an entity’s human-likeness and familiarity (not between anticipation and familiarity). The human-likeness/familiarity relationship is correctly discussed by the authors in the following sentence “Mori observed that the more human-like a technological object such as a robot is, the more familiar it seems; however, this is true only up to a certain point, at which even subtle deviations from human norms can make the object look disturbing”.

“Anticipation”, “perceptual difficulty”, as well as “increased sensitivity to defects in a human-like form”, mentioned in this paragraph, actually refer to different proposed explanations of the UV effect. First, “anticipation” plays a key role in the expectation violation explanation of the uncanny valley. The expectation violation explanation suggests that the negative response to an entity in the UV occurs when it violates the expectations that humans have for how a human-like entity should behave and interact based on their past experiences and knowledge of human behaviour. Second, the sentence “Perceptual difficulty in distinguishing between a human-like object and its human counterpart would produce negative affect” refers to the “categorization ambiguity” explanations of the UV. When an entity is difficult to categorize as either animate or inanimate because it possesses some but not all of the features of a living entity, this creates a sense of ambiguity, which can lead to discomfort and unease. Third, “Mori observed an increased sensitivity to defects in a human-like form”. The higher sensitivity to deviations from typical human norms for more human-like characters is part of the perceptual mismatch explanations, according to which the UV effect is caused by any perceptual mismatch in the digital entity’s appearance. In this case, as humans interact with entities that resemble humans to a greater degree, they become more attuned to the nuances of human behaviour and appearance. This heightened sensitivity can increase awareness of deviations (mismatches) from typical human norms. [If you wish to know more, please see Katsyri et al. (2015) “A review of empirical evidence on different uncanny valley hypotheses: support for perceptual mismatch as one road to the valley of eeriness”.]

Here I would suggest the authors keep it simple by just referring to the relationship between human-likeness and familiarity, without digging into the different explanations.

Furthermore, I would change the sentence “The hypothesis suggests that humanoid objects, including chatbots, which appear almost, but not exactly, like real human beings, may evoke strange or strangely familiar negative as well as positive feelings in observers” removing the reference to positive feeling, which creates confusion while talking about the UV effect.

Finally, as observed by the other reviewer in the first step of revisions, the meaning of “Additionally, there is a positive relationship between human likeness and knowledge”: is unclear. I would suggest the author to remove it.

At p. 3/13 the concept of utility is still unclear. First: “In real systems, chatbots […] are able to perform more accurate peripheral search and identification tasks while performing a central search”. I think there is a mistake here as it seems that are chatbots perfoming peripheral search and identification tasks. I guess the authors wanted to say that in real systems, given that chatbots are usually located peripherally, they require the users to perform a peripheral search while performing a central task (which is in line with the attention-utility issue discussed by McCrickard and colleagues).

The following phrase “Mimicked expressions of emotions are signals of high biological value” does not link with the one before and the one after and drift attention from the definition of the utility concept. I would suggest removing it.

The authors then discuss utility, but its definition is still hard to catch. Utility refers to a system's usefulness to its users or customers. What McCrickard and colleagues discuss in the context of peripheral system design, is that utility refers to the ways in which peripheral cues or messages help users accomplish their goals without requiring their full attention. Therefore, the sentence “Utility, like usability, can be evaluated as an aspect of human-computer interaction for the purpose of identifying aspects of this interaction that can be improved with the help of evaluation methods” still doesn’t capture the definition of utility.

Also, the difference in the conceptualization of utility between McCrickard and colleagues, and the one provided by the authors is unclear. What are the authors referring to when they talk about “individual gaps”? Once addressed this issue, I would place this section after the definition of utility for better clarity.

At p. 3/13 there is a “s” after the [25].

At p. 9/13 there is a missing in-text reference number (McCrickard et al.).

Reviewer #2: (No Response)

7. PLOS authors have the option to publish the peer review history of their article (what does this mean?). If published, this will include your full peer review and any attached files.

Reviewer #1: No

Reviewer #2: No

---

## [Author Response · Author response to Decision Letter 1]

13 Jun 2023

At the beginning, we would like to thank the Reviewer for valuable comments and tips. We appreciate the time spent and helpful remarks. Please find the answers and comments to the doubts presented in the review enclosed below.

Point 1: 

 At p. 2/13 “The ’social presence’ theory [11] suggests that the degree of social presence in a communication medium can affect the perceived level of social presence in the interaction”. This sentence is unclear as the authors are saying that “the degree of social presence […] can affect the level of social presence”, which is a tautology. The social presence theory suggests that the degree to which a medium (such as a chatbot) is designed to look and act more like a human, with features such as a human-like appearance, facial expressions, and gestures, positively impact the level of social presence perceived in the interaction. As a result, users may feel like they are engaging in a more natural and authentic conversation, as if they were talking to a real human instead of a machine.

Response 1: We agree with the Reviewer’s remarks and we have implemented their suggestions. We removed redundant sentence about tautology to. " the degree of social presence […] can affect the level of social presence". We made a correction in the text regarding this fragment in the comment. The new fragment of the text we have included in the work is: "The extent to which a medium, like a chatbot, is designed to resemble and behave like a human, incorporating elements such as a human-like appearance, facial expressions, and gestures, can significantly influence the perceived level of social presence during the interaction. Consequently, users may experience a heightened sense of engaging in a genuine and natural conversation, almost as if they were conversing with an actual human being rather than a machine." 

Point 2: 

At p. 2/13 “In fact, Mori [12] observed a degree of defect proneness in near-human forms an ”uncanny valley” in what otherwise follows from the relationship between anticipation and familiarity”. That is not entirely precise. The Uncanny Valley (UV) hypothesis refers to the relationship between an entity’s human-likeness and familiarity (not between anticipation and familiarity). The human-likeness/familiarity relationship is correctly discussed by the authors in the following sentence “Mori observed that the more human-like a technological object such as a robot is, the more familiar it seems; however, this is true only up to a certain point, at which even subtle deviations from human norms can make the object look disturbing”.

“Anticipation”, “perceptual difficulty”, as well as “increased sensitivity to defects in a human-like form”, mentioned in this paragraph, actually refer to different proposed explanations of the UV effect. First, “anticipation” plays a key role in the expectation violation explanation of the uncanny valley. The expectation violation explanation suggests that the negative response to an entity in the UV occurs when it violates the expectations that humans have for how a human-like entity should behave and interact based on their past experiences and knowledge of human behaviour. Second, the sentence “Perceptual difficulty in distinguishing between a human-like object and its human counterpart would produce negative affect” refers to the “categorization ambiguity” explanations of the UV. When an entity is difficult to categorize as either animate or inanimate because it possesses some but not all of the features of a living entity, this creates a sense of ambiguity, which can lead to discomfort and unease. Third, “Mori observed an increased sensitivity to defects in a human-like form”. The higher sensitivity to deviations from typical human norms for more human-like characters is part of the perceptual mismatch explanations, according to which the UV effect is caused by any perceptual mismatch in the digital entity’s appearance. In this case, as humans interact with entities that resemble humans to a greater degree, they become more attuned to the nuances of human behaviour and appearance. This heightened sensitivity can increase awareness of deviations (mismatches) from typical human norms. [If you wish to know more, please see Katsyri et al. (2015) “A review of empirical evidence on different uncanny valley hypotheses: support for perceptual mismatch as one road to the valley of eeriness”.]

Here I would suggest the authors keep it simple by just referring to the relationship between human-likeness and familiarity, without digging into the different explanations.

Furthermore, I would change the sentence “The hypothesis suggests that humanoid objects, including chatbots, which appear almost, but not exactly, like real human beings, may evoke strange or strangely familiar negative as well as positive feelings in observers” removing the reference to positive feeling, which creates confusion while talking about the UV effect.

Finally, as observed by the other reviewer in the first step of revisions, the meaning of “Additionally, there is a positive relationship between human likeness and knowledge”: is unclear. I would suggest the author to remove it.

Response 2: We agree with the Reviewer’s remarks and we have implemented their suggestions. We removed redundant sentence. We've modified the Uncanny Valley Theory section, which we've addressed in the section below:

"The extent to which a medium, like a chatbot, is designed to resemble and behave like a human, incorporating elements such as a human-like appearance, facial expressions, and gestures, can significantly influence the perceived level of social presence during the interaction. Consequently, users may experience a heightened sense of engaging in a genuine and natural conversation, almost as if they were conversing with an actual human being rather than a machine.

The visualization of chatbots and their social features should target users' expectations to ultimately avoid frustration and dissatisfaction. The effects of an electronic conversation on human behavior and the perceived level of anthropomorphism are a part of the broad issue of human attitudes towards humanoid technologies.

According to the theory proposed by Mori, the more a character resembles a human, the more it is accepted by us and evokes positive feelings. However, when a character becomes too realistic but still has subtle differences, such as unnatural movements or improper proportions, we experience a sense of unease or rejection in our minds. This is the moment when we enter the so-called Uncanny Valley.

Mori's theory suggests that the acceptance of artificial figures increases with their level of realism until a certain point, after which there is a sudden decline in acceptance. Only when a character reaches an exceptionally high level of realism, almost indistinguishable from a living human, does acceptance increase again.

The phenomenon of the Uncanny Valley has also been applied in the context of avatars with facial expressions. Irregularities in facial expression movements or inconsistencies with our expectations can make us feel uneasy and result in negative attitudes towards such avatars. Therefore, it is not surprising that the hypothesis regarding the Uncanny Valley has been adopted to explain the poor commercial success of some animated films in the media.

In a study conducted by Katsyri et al. (2015), existing research on people's reactions to artificial figures with varying degrees of realism was analyzed to investigate which hypotheses best explain this phenomenon. One hypothesis suggests that the feeling of unease in the Uncanny Valley arises from our social and cultural context. If artificial human-like figures are perceived as strange or inappropriate in our society, they can evoke negative emotions.

Study of Kao et al. (2019) finds that avatars with higher anthropomorphism led to higher player experience. Avatars with higher anthropomorphism also led players to identify more highly with their avatars. Independent of avatar type, we find avatar identification significantly promotes player experience. Players playing games doomed by little humanoidness. We will be more successful when the avatar is more a human."

Point 3: 

At p. 3/13 the concept of utility is still unclear. First: “In real systems, chatbots […] are able to perform more accurate peripheral search and identification tasks while performing a central search”. I think there is a mistake here as it seems that are chatbots perfoming peripheral search and identification tasks. I guess the authors wanted to say that in real systems, given that chatbots are usually located peripherally, they require the users to perform a peripheral search while performing a central task (which is in line with the attention-utility issue discussed by McCrickard and colleagues).

The following phrase “Mimicked expressions of emotions are signals of high biological value” does not link with the one before and the one after and drift attention from the definition of the utility concept. I would suggest removing it.

The authors then discuss utility, but its definition is still hard to catch. Utility refers to a system's usefulness to its users or customers. What McCrickard and colleagues discuss in the context of peripheral system design, is that utility refers to the ways in which peripheral cues or messages help users accomplish their goals without requiring their full attention. Therefore, the sentence “Utility, like usability, can be evaluated as an aspect of human-computer interaction for the purpose of identifying aspects of this interaction that can be improved with the help of evaluation methods” still doesn’t capture the definition of utility.

Also, the difference in the conceptualization of utility between McCrickard and colleagues, and the one provided by the authors is unclear. What are the authors referring to when they talk about “individual gaps”? Once addressed this issue, I would place this section after the definition of utility for better clarity.

Response 3: We agree with the Reviewer remarks and we have implemented their suggestions. Indeed, there was a mistake with subjects in the sentence, which disrupted the logical flow. Thank you for identifying that. We have worked on the corresponding excerpts to make them more clear. We have referred to the general understanding of utility and referred to the works of McCrickard and colleagues, adding an extra citation (McCrickard and Chewar) as well elaborated on what we understand by individual gaps in the context of utility. 

Point 4: 

 At p. 3/13 there is a “s” after the [25].

Response 4: 

The redundant letter has been removed.

Point 5: 

At p. 9/13 there is a missing in-text reference number (McCrickard et al.).

Response 5: Missing quote has been corrected.

---

## [Editor Report · Decision Letter 2]

20 Jun 2023

Impact of changes in chatbot's facial expressions on user attention and reaction time

PONE-D-22-20948R2

Dear Dr. Bortko,

We’re pleased to inform you that your manuscript has been judged scientifically suitable for publication and will be formally accepted for publication once it meets all outstanding technical requirements.

Kind regards,

Stefano Triberti, Ph.D.

Academic Editor

PLOS ONE
---

## [Editor Report · Acceptance letter]

19 Jul 2023

PONE-D-22-20948R2 

Impact of changes in chatbot's facial expressions on user attention and reaction time 

Dear Dr. Bortko:

I'm pleased to inform you that your manuscript has been deemed suitable for publication in PLOS ONE. Congratulations! Your manuscript is now with our production department. 

Kind regards, 

on behalf of

Dr. Stefano Triberti 

Academic Editor

PLOS ONE